# Dissecting the Activity of Catechins as Incomplete Aldose Reductase Differential Inhibitors through Kinetic and Computational Approaches

**DOI:** 10.3390/biology11091324

**Published:** 2022-09-06

**Authors:** Francesco Balestri, Giulio Poli, Lucia Piazza, Mario Cappiello, Roberta Moschini, Giovanni Signore, Tiziano Tuccinardi, Umberto Mura, Antonella Del Corso

**Affiliations:** 1Biochemistry Unit, Department of Biology, University of Pisa, Via S. Zeno, 51, 56123 Pisa, Italy; 2Interdepartmental Research Center Nutrafood “Nutraceuticals and Food for Health”, University of Pisa, 56124 Pisa, Italy; 3Department of Pharmacy, University of Pisa, Via Bonanno Pisano 12, 56126 Pisa, Italy

**Keywords:** aldose reductase, catechins, diabetic complications, differential inhibitors, incomplete inhibition

## Abstract

**Simple Summary:**

The increased glucose levels occurring in diabetes lead to several metabolic alterations responsible for the onset of the so-called diabetic complications, which include nephropathies, neuropathies, retinopathies, and cataract. An increased flux of glucose through the polyol pathway is considered the most relevant among these alterations. For this reason, the block of the polyol pathway, through the inhibition of the enzyme aldose reductase, is considered a valuable strategy to impair the onset of diabetic complications. However, aldose reductase also exerts a beneficial effect inside cells, since it can remove toxic aldehydes. Thus, to ameliorate the outcome of the use of aldose reductase inhibitors, the use of “differential inhibitors” has been proposed. These inhibitors should block the catalytic activity depending on the substrate the enzyme is working on, thus preserving the detoxifying action of the enzyme. In this work, derivatives of catechins are analyzed to evaluate their inhibitory action on aldose reductase. The study was conducted both in vitro on the isolated enzyme and in silico through a computational approach. Results demonstrated that gallocatechin gallate and catechin gallate act as differential inhibitors and that this action may be linked to an incomplete inhibitory effect.

**Abstract:**

The inhibition of aldose reductase is considered as a strategy to counteract the onset of both diabetic complications, upon the block of glucose conversion in the polyol pathway, and inflammation, upon the block of 3-glutathionyl-4-hydroxynonenal reduction. To ameliorate the outcome of aldose reductase inhibition, minimizing the interference with the detoxifying role of the enzyme when acting on toxic aldehydes, “differential inhibitors”, i.e., molecules able to inhibit the enzyme depending on the substrate the enzyme is working on, has been proposed. Here we report the characterization of different catechin derivatives as aldose reductase differential inhibitors. The study, conducted through both a kinetic and a computational approach, highlights structural constraints of catechin derivatives relevant in order to affect aldose reductase activity. Gallocatechin gallate and catechin gallate emerged as differential inhibitors of aldose reductase able to preferentially affect aldoses and 3-glutathionyl-4-hydroxynonenal reduction with respect to 4-hydroxynonenal reduction. Moreover, the results highlight how, in the case of aldose reductase, a substrate may affect not only the model of action of an inhibitor, but also the degree of incompleteness of the inhibitory action, thus contributing to differential inhibitory phenomena.

## 1. Introduction

Aldose reductase (E.C. 1.1.1.21; AKR1B1) is a NADPH-dependent oxidoreductase, belonging to the aldo-keto reductase superfamily [1]. The involvement of AKR1B1 in the aetiology and progression of pathological states related to diabetes (the so-called “diabetic complications”) is well documented [2,3] and represents the rational basis for the strong experimental effort, ongoing since decades, in attempting to inhibit the enzyme [4,5,6,7,8]. The enzyme is metabolically located in the polyol pathway in which it catalyzes the first step, i.e., the conversion of glucose to sorbitol. A second step, catalyzed by a NAD-dependent sorbitol dehydrogenase, allows the conversion of sorbitol to fructose and completes the pathway. AKR1B1 is responsible for the increased flux of glucose observed through the polyol pathway in hyperglycemic conditions. This alteration induces at the same time osmotic, oxidative, and glycative insults which lead to the onset of diabetic complications [2]. The assessed multi-specificity of AKR1B1, which is able to act with different effectiveness on structurally quite different substrates [9,10,11,12], evokes a conflictual view of the enzyme inhibition. In fact, AKR1B1, beside its ability to reduce glucose and other aldoses in hyperglycemic conditions, leading to cells damage is able to reduce a number of cytotoxic aldehydes, thus exerting a detoxifying role. The case of the 3-glutathionyl-4-hydroxynonanal adduct (GSHNE) deserves particular attention. In fact, it has been suggested that GSHNE, through the generation of the corresponding alcohol, is involved in the NF-kB mediated inflammatory response [13,14]. Thus, the inhibition of AKR1B1 has been considered as a strategy to elicit an anti-inflammatory response, as observed in several cell types and experimental models exposed to different inflammatory stimuli [15,16,17,18]. On the basis of the above considerations, the so-called “differential inhibition” approach, aimed to inhibit the transformation of glucose and GSHNE, with a null or minimum impact on the removal of toxic aldehydes, has been proposed [19,20]. This approach would ameliorate the outcome of AKR1B1 inhibition, preserving the anti-inflammatory and detoxifying action of the enzyme, which would be beneficial for diabetic patients. The not-permissive features of AKR1B1, as deduced by its specificity constants toward substrates of the same structural class [11] and its ability to establish peculiar interactions with different substrates depending on their structural features [21], support the feasibility of the differential inhibition approach. The ability of synthetic and natural molecules to act as aldose reductase differential inhibitors (ARDIs) has been reported. In particular, derivatives of pyrazolo [1,5-α]pyrimidine endowed with a carboxylic group have been shown to inhibit the reduction of aldoses and GSHNE more efficiently than the reduction of HNE [22]. Triterpenoid saponins isolated from Zolfino bean have been identified as inhibitors of AKR1B1; among them, soyasaponin Bb resulted in affecting enzyme activity depending on the substrate, thus acting as ARDI [23]. Recently, green tea has been shown as a potential source of ARDIs, and epigallocatechin gallate (EGCG), epigallocatechin (EGC), and gallic acid (GA) were characterized for their ability to differentially inhibit AKR1B1 [21]. While EGC inhibits AKR1B1 irrespective of the used substrate, EGCG and GA act as ARDIs, even though not complete. In fact, both these compounds affected HNE reduction, but to a significantly less extent with respect to the aldose reduction.

Here, we report the characterization of additional catechin derivatives as potential ARDIs. Our results increase the number of molecules exerting a substrate-dependent inhibitory action of AKR1B1 and highlight that the occurrence of an incomplete inhibition may contribute to a differential inhibitory effect.

## 2. Materials and Methods

### 2.1. Materials

Gallocatechin gallate (GCG; HPLC purity > 98%), catechin gallate (CG; HPLC purity > 98%), epicatechin gallate (ECG; HPLC purity > 97.5%), catechin (C; HPLC purity > 99%), gallocatechin (GC; HPLC purity > 98%), and epicatechin (EC; HPLC purity > 99%) were from Extrasynthese (Genay Cedex, France); NADPH and L-idose were from Carbosynth (Compton, England). Bovine serum albumin (BSA), D,L-dithiothreitol (DTT), D,L-glyceraldehyde (GAL), EDTA, glycerol and glutathione (GSH) were purchased from Sigma-Aldrich (Saint Louis, MO, USA); ammonium sulfate, sodium phosphate monobasic were from J.T. Baker (Phillipsburg, NJ, USA); YM10 ultrafiltration membranes were from Merck-Millipore (Darmstadt, Germany). All other chemicals were of a reagent grade.

### 2.2. Determination of AKR1B1 Activity

The AKR1B1 activity was determined at 37 °C, as previously described [24], monitoring the decrease in absorbance at 340 nm linked to NADPH oxidation (ε_340_ = 6.22 mM^−1^·cm^−1^) using a Biochrom Libra S60 spectrophotometer. The standard assay mixture (0.7 mL) contained 0.25 M sodium phosphate buffer pH 6.8, 0.18 mM NADPH, 0.4 M ammonium sulfate, 0.5 mM EDTA, and 4.7 mM GAL. One unit of enzyme activity is the amount of enzyme that catalyzes the conversion of 1 µmol of substrate/min in the above assay conditions. The same assay conditions were adopted in the inhibition studies with L-idose, HNE, or GSHNE as substrates.

IC_50_ values were determined by non-linear regression analysis using GraphPad Prism version 7.04 (GraphPad Software, San Diego, CA, USA). Values were obtained from at least six different inhibitors concentrations (each assayed at least in triplicate) and are reported with their 95% Confidence Limits (CLs). Inhibitors were dissolved either in water (GCG, CG, ECG) or in 40% ethanol (C, GC, EC). The final concentration of ethanol in the assay was never higher than 1%. The presence of ethanol did not affect the inhibitory ability of GCG, CG, and ECG. The percentage of inhibition was calculated referring to an assay performed using the same solvent concentration in the absence of inhibitors.

### 2.3. Purification of Human Recombinant AKR1B1

The human recombinant AKR1B1 was expressed and purified to electrophoretic homogeneity, as previously described [25]. The specific activity of the purified enzyme was 5.3 U/mg. The purified enzyme was stored at −80 °C in a 10 mM sodium phosphate buffer pH 7.0 containing 2 mM DTT and 30% (*w*/*v*) glycerol. Before use, AKR1B1 was extensively dialyzed against 10 mM sodium phosphate buffer pH 7.0.

### 2.4. Kinetic Analysis

The kinetic parameters *^app^k_cat_* and *^app^K_M_* were evaluated by non-linear regression analysis of rate measurements vs. substrate concentration according to Michaelis-Menten equation. The analysis was performed through GraphPad Prism 7.04 software by a non-linear “Robust Regression” analysis in which each point is individually weighted through iterative weighing of the smallest squares [26]. The same approach was adopted to analyze the dependence of *^app^k_cat_*, *^app^K_M_*, and *^app^K_M_*/*^app^k_cat_* from the concentration of the inhibitors [I], making use of previously derived equations [27] (see text).

### 2.5. Molecular Docking

The X-ray structure of human AKR1B1 in complex with D-glyceraldehyde and the NADP cofactor (PDB code 3V36) [28], as well as the co-crystal structure of human AKR1B1 in complex with a nitrofuryl-oxadiazol inhibitor (PDB code 2IKH) [29], were downloaded from the Protein Data Bank [30]. Molecular docking calculations were performed with GOLD 5.1 (CCDC Software Ltd., Cambridge, UK) [31] with the Piecewise Linear Potential (PLP) fitness function; both the two X-ray structures and the previously predicted protein–substrate complexes (i.e., AKR1B1-L-idose, AKR1B1-HNE, and AKR1B1-GSHNE) [21] were employed as the protein. The region of interest for the docking calculations included all the residues, which stayed within 10 Å from the bound molecule in either the corresponding X-ray structure or the predicted protein–substrate complexes. The ligands were subjected to 100 genetic algorithm runs, in which the “allow early termination” option was deactivated, while the possibility for the ligand to flip ring corners was activated. All other settings, including the root-mean-squared deviation (RMSD) threshold for pose clustering (2.0 Å), were left as their defaults. The best docked conformation of each cluster of solution was considered in each docking study. When protein–substrate complexes were used as the receptor, the ten top-scored clusters of solutions in which the ligand formed at least one H-bond with the substrate were selected as the best clusters of solutions, which were considered for further analyses.

### 2.6. Molecular Dynamics Simulations

Molecular dynamics (MD) simulations were performed with AMBER 20 [32] using the ff14SB force field. General Amber force field (GAFF) parameters were used for the ligands, the substrates, and the cofactor, whose partial charges were calculated with the AM1-BCC method as implemented in the Antechamber suite of AMBER 20 (University of California, San Francisco, CA, USA). All simulations were performed using particle mesh Ewald electrostatics, a cut-off of 10 Å for non-bonded interactions, and periodic boundary conditions. All bonds involving hydrogens were kept rigid using the SHAKE algorithm, and a simulation step of 2.0 fs was thus employed. Each analyzed complex was placed in a rectangular parallelepiped water box, using the TIP3P explicit solvent model for water, and solvated with a 15 Å water cap. Sodium ions were added as counterions for the neutralization of the systems. Initially, the systems were energy minimized through two minimization steps, including 5000 cycles of steepest descent followed by the conjugate gradient algorithm, until a convergence of 0.05 kcal/Å mol. In the first step, the protein was maintained rigid with a position restraint of 100 kcal/mol·Å^2^, thus minimizing only the positions of the water molecules. In the second step, the whole system was energy minimized by applying a harmonic potential of 10 kcal/mol·Å^2^ only to the protein α-carbons. The minimized complexes were then used as the starting point for the MD simulations, applying a protocol adapted from previous studies [33,34]. A 0.5 ns constant-volume simulation, in which the temperature of the system was raised from 0 to 300 K, was initially performed. The system was then equilibrated with 3 ns of constant-pressure simulation, keeping a constant temperature of 300 K with the use of Langevin thermostat. Finally, 46.5 ns of MD simulations with constant pressure and temperature conditions were performed for a total of 50 ns of simulation. In all MD steps, the harmonic potential of 10 kcal/mol·Å^2^ on the protein α-carbons was maintained.

### 2.7. Binding Energy Evaluations

Binding free energy evaluations were performed using AMBER 20 (University of California, San Francisco, CA, USA), as previously performed [35,36]. The trajectories extracted from the last 30 ns of each simulation were used for the calculation, for a total of 300 snapshots (at time intervals of 100 ps). Van der Waals, electrostatic, and internal interactions were calculated with the SANDER module of AMBER 20, whereas the Poisson−Boltzman method was employed to estimate polar energies through the MM-PBSA module of AMBER 20. Gas and water phases were represented using dielectric constants of 1 and 80, respectively, while nonpolar energies were calculated with the MOLSURF program. The entropic term was considered as approximately constant in the comparison of the ligand–protein energetic interactions.

### 2.8. Other Methods

Protein concentration was determined by the Coomassie blue staining method [37], using BSA as a standard protein. HNE was synthetized as previously described [38]. GSHNE was prepared as previously described [38] by incubating GSH and HNE (1.5:1 molar ratio) and monitoring the time course of GSH consumption [39].

## 3. Results

### 3.1. AKR1B1 Inhibition by Catechins: Evidence of Differential Inhibition

The ability of EGCG to act as ARDI in the reduction of L-idose, with respect to HNE and GSHNE [21], induced the evaluation of possible ARDI features in different catechin analogues. Thus, the inhibitory ability of GCG, CG, ECG, C, GC, and EC was investigated. The structures of these compounds are reported in Figure 1. Catechin, EC, and GC resulted as essentially ineffective as AKR1B1 inhibitors; a residual activity of approximately 70% was measured in the presence of 0.8 mM of these compounds and they were not further investigated. The residual reductase activity measured for different substrates in the presence of different concentrations of GCG, CG, and ECG is reported in Figure 2. Assays were performed using concentrations in the millimolar range for the sugar substrate and in the micromolar range for GSHNE and HNE. The results revealed, for these compounds, an appreciable differential inhibitory action. GCG inhibited the reduction of L-idose, used as substrate mimicking glucose, with a IC_50_ of 13 µM (11–17, 95% CLs) and affected both HNE and GSHNE reduction with comparable IC_50_ values of 70 µM (62–79, 95% CLs) and 61 µM (49–76, 95% CLs), respectively. A higher absolute inhibitory power was observed for CG, with respect to GCG, for all the tested substrates (Figure 2B). Concerning the ARDI ability, the IC_50_ value for the reduction of L-idose, (7.0 µM; 6–8, 95% CLs) resulted as approximately four times lower than those measured for HNE and GSHNE reduction, which were essentially identical (26 µM, 23–30, 95% CLs and 28 µM, 23–33, 95% CLs, respectively). Thus, both GCG and CG display a comparable differential inhibition of the sugar reduction with respect to both HNE and GSHNE reduction. Conversely, EGCG apparently preserved only HNE reduction [21].

Finally, ECG displayed a lower inhibitory power with respect to both GCG and CG. The obtained IC_50_ values for the reduction of L-idose, HNE, and GSHNE were 67 µM (53–84, 95% CLs), 257 µM (216–311, 95% CLs), and 143 µM (124–165, 95% CLs), respectively.

### 3.2. Inhibition Kinetic Analysis 

The inhibitory features of the tested catechins were furthered by a detailed kinetic analysis using L-idose, HNE, and GSHNE as substrates. Rate measurements acquired at different inhibitor concentrations are shown as plots of *v*_0_ versus substrate concentrations (Figure 3).

The kinetic analysis to evaluate the apparent kinetic parameters *^app^K_M_* and *^app^k_cat_* were performed by non-linear regression analysis; the resulting secondary plots of *^app^K_M_*, *^app^k_cat_* and *^app^K_M_*/*^app^k_cat_* versus the inhibitor concentration are shown in Figure 4, Figure 5 and Figure 6.

It is evident that, for all the substrates, both *^app^k_cat_* and *^app^K_M_* decrease with the increase of the inhibitor concentration. This, referring to the classical reaction scheme of Figure 7, in which *k*_+4_ is imposed equal to zero, would be compatible with the inhibitor targeting the enzyme–substrate complex more efficiently than the free enzyme. Thus, the apparent dissociation constant of the enzyme inhibitor complex *K_i_* (i.e., the intercept on the abscissa of the replot *^app^K_M_*/*^app^k_cat_*) would be higher than *K’_i_* (i.e., the dissociation constant of the ternary enzyme inhibitor substrate complex, EIS). Indeed, in the case of an uncompetitive inhibition (i.e., extremely high value for *K_i_*), the slope of *^app^K_M_*/*^app^k_cat_* versus [I] approaches to zero. This is the case of L-idose reduction in the presence of all the tested compounds. On the contrary, in the case of CG and GCG for HNE reduction and at least in the case of CG for GSHNE reduction, a decrease of the *^app^K_M_*/*^app^k_cat_* versus [I] is observed. Moreover, the observed trend could hardly be interpolated with a straight line.

This behavior has also been described for the effect of EGCG on HNE reduction and rationalized through an inhibition model characterized by an incomplete inhibition [27], in which the possibility for the ternary complex EIS to evolve to products is considered (*k*_+4_ > 0) (Figure 7).

The hyperbolic kinetic equation for an incomplete mixed type of inhibition analyzed with both branches of product formation at the steady state and with the inhibitor binding at equilibrium, was defined using the apparent kinetic parameters *^app^K_M_* and *^app^k_cat_* (Equations (1)–(3), see ref [27] for further details).
(1)kappcat=Ki*k+2+k+4[I]Ki*+[I]
(2)$$appK_{M}=\frac{K_{M}K_{i}^{*}+\left(\frac{K_{i}^{*}}{K_{i}}K_{M}+\frac{k_{+4}}{k_{+1}}\right)\left[I\right]+\frac{k_{+4}}{K_{i}k_{+1}}{\left[I\right]}^{2}}{K_{i}^{*}+\left[I\right]}\;$$
(3)$$\frac{appK_{M}}{appk_{cat}}=\frac{\;K_{M}K_{i}^{*}+\left(\frac{K_{i}^{*}}{K_{i}}K_{M}+\frac{k_{+4}}{k_{+1}}\right)\left[I\right]+\frac{k_{+4}}{K_{i}k_{+1}}{\left[I\right]}^{2}}{K_{i}^{*}k_{+2}+k_{+4}\left[I\right]}$$

In the case of an incomplete, uncompetitive inhibition, the following relation can be derived: (4)KappMkappcat=Ki*KM+k+4k+1[I]Ki*k+2+k+4[I]
where the term $K_{i}^{*}$ refers to the dissociation of the ternary complex EIS perturbed by the evolution of the complex to products and is defined as:(5)$$K_{i}^{*}=\frac{k_{-3}+k_{+4}}{k_{+3}}$$

Thus, *^app^k_cat_* and *^app^K_M_* values emerging from non-linear regression analysis of primary data (Figure 3) were analyzed through non-linear regression using Equations (1) and (2), respectively. To evaluate the apparent dissociation constant of EIS (*K_i_^*^*) and the kinetic constant of the ternary complex (*k*_+4_), the starting values of *k_cat_* (i.e., *k*_+2_) for different substrates in the absence of the inhibitor were imposed for the analysis of Equation (1) as 195 (195 ± 6) min^−1^, 78 (78 ± 4) min^−1^, and 109 (109 ± 8) min^−1^ for L-idose, HNE, and GSHNE, respectively. To determine the *ES* dissociation constant (*K_i_*), whose value was restricted in Equation (2) being >0, the starting values of *K_M_* for the substrates L-idose, HNE, and GSHNE were 4260 (4260 ± 300) µM, 51 (51 ± 5) µM, and 120 (120 ± 16) µM, respectively. The above starting values of *k_cat_* and *K_M_* derived from at least 20 independent experiments in which kinetic parameters were measured in the absence of inhibitors. Equations (2) and (3) could be simplified, having verified the expected extremely low values of *k*_+4_*/k*_−1_, by imposing this ratio equals to zero. Finally, when *K_i_* values far exceed *K_i_^*^* values (*K_i_*/*K_i_^*^* at least > 100), the *^app^K_M_*/*^app^k_cat_* versus [I] data were interpolated by Equation (4) as an incomplete, uncompetitive inhibition. With such an approach [27], the emerged kinetic parameters reported in Table 1 give rise to the curves fitting of the experimental results of Figure 4, Figure 5 and Figure 6. The *k*_+4_ values are reported in Table 1 as the average of values emerging from *^app^K_M_*/*^app^k_cat_* versus [I] (Equation (3) or Equation (4)) and 1/*^app^k_cat_* versus [I] (Equation (1)) plots. An essentially complete inhibition occurred for all the inhibitors in the case of L-idose reduction (*k*_+4_ ranging from approximately 0.7% to 2.9% of *k*_+2_). Conversely, an incomplete inhibition by CG, GCC, and ECG, respectively, was observed for HNE reduction, with *k*_+4_ values resulting approximately 6, 12, and 9% of *k*_+2_. Additionally, for GSHNE reduction, the inhibition appears not complete, with *k*_+4_ values resulting as approximately 4, 6, and 14% of *k*_+2_ for CG, GCG, and ECG, respectively.

### 3.3. Computational Study of AKR1B1 Inhibition

Computational studies including docking, MD simulations, and binding free energy evaluations were employed in the attempt to rationalize the kinetic evidence about AKR1B1 inhibition. ECG appeared significantly less effective than the other tested inhibitors (*K_i_* and *K^*^_i_* one order of magnitude higher with respect to CG and GCG). On the other hand, ECG was the one more closely behaving as the previously studied EGCG [21] and even more active, especially on HNE and GSHNE. Thus, the computational study was limited to GCG and CG, whose catechin scaffold has never been analyzed before in terms of AKR1B1 interaction. Due to the extremely high structural similarity among GCG and CG, which differ only for a single phenolic group, GCG was used as a representative compound for the modeling studies.

According to the kinetic analysis (Table 1), we first aimed at predicting the bioactive conformation of GCG into AKR1B1 binding site in the presence of the three different substrates. In order to evaluate the potential corresponding ESI complex with L-idose, GCG was docked into the L-idose-bound AKR1B1 structure predicted in our previous work [21]. The same docking protocol based on GOLD software was applied and the ten best clusters of solutions were considered for further studies (see Materials and Methods for details). The best docked pose of each cluster was evaluated through the same 50 ns MD simulation protocol previously employed. The stability of the ten corresponding ligand–protein–substrate complexes was analyzed in terms of average root-mean-squared deviation (RMSD) of ligand disposition during the whole MD, with respect to its initial docking pose, and inhibitor–ligand binding free energy (between the inhibitor and the enzyme–substrate complex), which was calculated using the Molecular-Mechanics–Poisson-Boltzmann Surface Area (MM-PBSA) approach (see Materials and Methods for details). As reported in Appendix A, complex **1** showed to be the most reliable, since it was associated to a binding free energy (ΔPBSA = −25.2 kcal/mol) with an energy gain of at least 7 kcal/mol compared with those estimated for the other complexes and showed one of the lowest values of average ligand RMSD (corresponding to 2.26 Å). As shown in Figure 8, GCG was predicted to bind to the AKR1B1-L-idose complex placing the gallic acid (GA) moiety in the amphiphilic cavity constituted by W219, C298, A299, L300, L301, and S302, which was previously predicted to be occupied by GA itself in the AKR1B1-L-idose-GA complex [21]. In fact, the GA moiety of GCG formed a strong π-π stacking with W219 and lipophilic interactions with L301, being sandwiched between the two residues, and anchored the inhibitor to the enzyme through three different H-bonds. In particular, two H-bonds were formed with the backbone nitrogen of A299 and L300, while the third one was established with the C6-hydroxyl group of the substrate and was maintained for more than 95% of the whole MD simulation. The polyphenolic ring connected to the bicyclic core of the ligand formed another direct H-bond with the C5-hydroxyl group of the substrate and a water-bridged H-bond with its C4-OH group. Moreover, a second water molecule mediated an additional H-bond interaction among ligand, substrate, and the indole moiety of W20. Finally, the central scaffold of the ligand, which laid against S302, formed another stable H-bond with the backbone oxygen of G128 that contributed to stabilize the binding pose of the molecule. The network of interactions between the substrate and the ligand, as well as the steric hindrance produced by the ligand, which virtually clamped the terminal portion of the substrate, are believed to lock L-idose within the enzyme binding site, thus inhibiting substrate turnover.

Data coming from the kinetic analysis of L-idose reduction (Table 1) revealed that both GCG and CG were also able to interact with the free enzyme. Therefore, we aimed at predicting the binding disposition of GCG in its binary complex with AKR1B1. For this purpose, based on the definition of the EI complex, we considered two possible different starting conformations of the enzyme, compatible with the interaction of both L-idose and the hydrophobic aldehyde substrates. Therefore, we docked GCG into the X-ray structure of AKR1B1 in complex with D-glyceraldehyde (PDB code 3V36) [28], previously used for predicting the AKR1B1-L-idose complex. In this case, the docking protocol generated only nine clusters of poses, which were analyzed by applying the same MD protocol and evaluating ligand RMSDs and ligand–protein binding free energies. As shown in Appendix A, similar binding energies (between −22.8 and −24.2 kcal/mol) were estimated for a few predicted AKR1B1-GCG complexes, i.e., complexes **2**, **6**, and **8**. However, the binding poses of the ligand in these three complexes were found to converge during the MDs. In particular, the binding pose of GCG in complexes **6** and **8** converged into the binding pose assumed in complex **2**, which was associated to both the lowest average ligand RMSD (0.88 Å) and the best binding free energy (−24.2 kcal/mol) among all analyzed complexes. Considering this, and that the binding energy gain associated to complex **2** outperformed at least 4 kcal/mol, ND those calculated for the other six non-converging complexes (**1**, **3–5**, **7** and **9**), complex **2** resulted as the most reliable one. As expected, due to the absence of the substrate, GCG was predicted to bind the enzyme by fully occupying its catalytic site. In fact, the inhibitor protruded within the anion binding pocket forming an H-bond with the catalytically relevant H110, a π-π stacking with W20 and additional hydrophobic interactions with W79 (Figure 9A). However, most of the predicted ligand–protein interactions anchoring the ligand to the enzyme involved the outer portion of AKR1B1 binding site. In fact, the ligand still occupied the amphiphilic pocket with the aromatic ring directly connected to its bicyclic core, forming two H-bonds with the backbone nitrogen of A299 and S302, as well as a π-π stacking with W219. Moreover, the GA moiety of the inhibitor showed an additional H-bond with the side chain of S302 and an optimal π-π stacking with F122. Compared to the ESI complex in which GCG interacts both with the protein and L-idose (Figure 8), the ligand was predicted to form a lower number of key anchoring H-bonds (four instead of six), but it could still extensively interact with the outer portion of the binding site, and it was also able to reach the inner catalytic cavity, thus stably occluding its access to the substrate. These data were consistent with both the ability of GCG to inhibit the AKR1B1-catalyzed reduction of L-idose by forming an EI complex and the reduced inhibitory potency observed (*K_i_* = 71 μM) with respect to that associated to the formation of the ESI complex (*K’_i_* = 7.1 μM). Moreover, the higher *K_i_* observed for the CG-mediated inhibition of L-idose reduction (*K_i_* = 246 μM) could be explained considering that CG lacks one of the two phenolic groups of GCG involved in the H-bonds with the backbone of A299 and S302 in the EI complex (Figure 9A). Therefore, CG would lack one of these two interactions, which would determine a reduced affinity for the enzyme alone.

A second approach to generate an EI complex was then devised. In our previous work, we hypothesized that HNE and GSHNE could bind to AKR1B1 by inducing the rotation of L300 toward W219, thus opening a mainly hydrophobic cavity, adjacent to the anion binding pocket, known as specificity pocket, similar to what occurs in the presence of many different ARIs [40]. By following this hypothesis, we were able to predict reliable AKR1B1-HNE and AKR1B1-GSHNE complexes in which the hydrophobic tail of HNE and the alkyl portion of GSHNE were optimally located within the specificity pocket of the enzyme. Interestingly, the conformational change associated to the opening of the specificity pocket involved residues L300, L301, and S302 included in the amphiphilic pocket, in which GCG appeared to form most of the interactions stabilizing the EI complex (Figure 9A). This would suggest that a binary EI complex generated with the enzyme adopting such conformation, with the open specificity pocket, might show a reduced stability. In order to test this hypothesis, GCG was docked into the X-ray structure derived from AKR1B1 in complex with a nitrofuryl-oxadiazol inhibitor occupying the specificity pocket (PDB code 2IKH) [29], which we previously used to predict our AKR1B1-HNE/GSHNE complexes. Only four clusters of poses were generated and thus only four ligand–protein complexes were subjected to MD simulations and analyzed (Appendix A). Complex **4** appeared to be the most reliable, presenting a binding free energy of −19.4 kcal/mol that was at least 5.4 kcal/mol lower than those calculated for the other complexes. As shown in Figure 9B, GCG still appeared able to reach the anion binding pocket, succeeding in forming two H-bonds with the amide oxygen of the cofactor and the indole moiety of W111. However, no effective interactions between the ligand and the amphiphilic pocket residues were observed. Precisely, the aromatic ring of the ligand directly connected to its bicyclic core could not be properly placed into the amphiphilic pocket anymore, since the side chain of L300 was closer to W219 while L301 and S302 were slightly pushed away from it. This determined the loss of the two H-bonds between the ligand and the backbone of A299 and S302, as well as the π-π stacking with W219. Moreover, the GA moiety of the ligand was far from S302 and F122, thus losing another H-bond and a π-π stacking with the former and the latter residue, respectively. Even the interactions with W20 and W79 appeared to be attenuated due to the unsuitable disposition of the ligand core. The lack of all these interactions, which were instead predicted for GCG interacting with the “closed specificity pocket conformation” of the enzyme (Figure 9A), account for the remarkably reduced stability of this complex. Accordingly, GCG showed significantly higher RMSD and binding free energy values when interacting with the open (Figure 9B) specificity pocket conformation (Appendix A; RMSD = 3.22 Å; ΔPBSA = −19.4 kcal/mol) compared to the closed (Figure 9A) specificity pocket conformation of the enzyme (Appendix A; RMSD = 0.88 Å; ΔPBSA = −24.2 kcal/mol). These results may provide a rationale for explaining the lack of inhibitory activity observed for GCG and CG upon formation of EI complexes in the presence of HNE and GSHNE (see the Discussion section).

At this point, we aimed at predicting the potential binding mode of GCG in complex with the HNE- and GSHNE-bound enzyme. We thus docked GCG into the AKR1B1-HNE and AKR1B1-GSHNE complexes predicted in our previous work [21] and analyzed, through MD simulations, the binding modes representing the best clusters of solutions by applying the same computational protocols. The most reliable AKR1B1-HNE-GCG complex analyzed was found to be complex **1**, presenting a binding free energy (−23.3 kcal/mol) that outperformed, by at least 4.5 kcal/mol, those calculated for the other complexes (Appendix A), with the exception of complex **6** (−22.7 kcal/mol), which in fact rapidly converged into complex **1** during the MD, thus further supporting its reliability. Figure 10A shows that the presence of the bound HNE allowed GCG to effectively interact with the enzyme, both providing key anchoring points for the inhibitor and further modulating the conformation of the amphiphilic pocket. Precisely, the alkyl chain of HNE pushed the side chain of L300 toward the solvent much more pronouncedly with respect to the disposition in the absence of HNE (Figure 9B). This allowed the GA moiety of the ligand to be placed again into the amphiphilic pocket, and to form two H-bonds with the backbone oxygen of V297 and the backbone nitrogen of A299, as well as a π-π stacking with W219 and lipophilic interactions with L300. Moreover, the ligand formed two different H-bonds with the hydroxyl group of HNE and a further H-bond with the indole moiety of W20, thus appearing to lock the substrate into AKR1B1 catalytic site. Finally, the central core of the ligand mainly showed hydrophobic interactions with F122 and L300.

The analysis of the ten potential AKR1B1-GSHNE-GCG complexes subjected to the MD protocol clearly suggested complex **5** as the most reliable one, showing the lowest value of average ligand RMSD value (2.04 Å) and a binding free energy (−25.5 kcal/mol) outperforming those estimated for the other complexes of at least 9 kcal/mol (Appendix A). As shown in Figure 10B, the bound substrate, which incorporates the same alkylic chain present in HNE, similarly modulated the shape of the amphiphilic pocket, whose residues showed the same conformation observed in the AKR1B1-HNE-GCG complex (Figure 10A). Consistently, the GA moiety of the ligand was similarly placed within this pocket, again forming a π-π stacking with W219, lipophilic interactions with L300, and an H-bond with the backbone nitrogen of A299. Two very stable H-bonds were observed between the ligand and the substrate, one involving the hydroxyl group of GSHNE and the other one established with the carbonyl group of its cysteine-derived moiety; moreover, two additional H-bonds were formed between two of the phenolic groups of the ligand (one belonging to its bicyclic core and one belonging to the connected phenyl ring) and the two carboxylic groups of GSHNE, although less stably maintained. Finally, the ligand formed a π-π stacking with F122. Despite the bigger size of GSHNE compared to HNE and L-idose, the predicted AKR1B1-GSHNE-GCG complex, in which the ligand formed interactions with all accessible moieties of the substrate, supported the hypothesis that GCG can also lock GSHNE within the enzyme binding site, hampering its turnover and its catalytic reduction.

Overall, the results herein obtained provide a rationale for explaining the remarkable inhibitory activity of GCG (and its analogue CG) observed upon formation of ESI complexes, meanwhile supporting the possibility of generation of binary inhibitor–enzyme complexes selectively obtained in presence of L-idose, which were indeed kinetically revealed. With the aim of deriving also possible clues for rationalizing the complete inhibition of L-idose in contrast to the incomplete inhibition of HNE and GSHNE produced by GCG, additional binding free energy evaluations were performed. In particular, we calculated the binding affinity of L-idose in the ESI complex in presence of GCG (Figure 8), based on the MD simulation of the complex. The obtained binding free energy value (−22.9 kcal/mol) was found to be 5.5 kcal/mol lower than the binding affinity estimated for L-idose bound to AKR1B1 in the absence of GCG (−17.4 kcal/mol) [21]. In particular, the presence of GCG determined a 32% increase in the binding affinity of L-idose for the complex, thus supporting the hypothesis that the inhibitor can efficiently lock the substrate within the enzyme binding site, hampering its turnover and catalytic reduction. The same analysis was then performed for the other two AKR1B1 substrates, by evaluating the binding free energy of HNE and GSHNE in the corresponding ESI complexes in presence of GCG (Figure 10A,B). As for L-idose, the binding affinities of both HNE (−20.6 kcal/mol) and GSHNE (−33.6 kcal/mol) were found to be lower than those estimated in absence of GCG (−18.6 and −30.3 kcal/mol, respectively) [21]. However, the increase in binding free energy estimated for the two substrates due to the presence of GCG (about 11%) was found to be approximately one third of that calculated for L-idose. These results suggest that, although GCG is able to inhibit the turnover of HNE and GSHNE, this effect is less pronounced than that estimated for L-idose, whose binding affinity is much more affected by the presence of GCG. This is consistent with the kinetic analysis demonstrating that only an incomplete inhibition of HNE and GSHNE is produced by GCG, in contrast to the complete inhibition observed for L-idose (Table 1). 

## 4. Discussion

The present study on the ability of catechins to inhibit AKR1B1 reveals some peculiar aspects of this class of compounds concerning the specificity of their action towards different substrates. As previously observed for EGCG [21,27], our results confirmed the relevant role of the galloyl moiety esterifying the hydroxyl group at C3 position of the molecule in order to exploit the inhibitory action. In fact, EC, EGC, C, and GC (Figure 1) did not inhibit AKR1B1. This galloyl moiety is, indeed, strongly involved in the stabilization of the ternary complexes with all the three substrates (Figure 8 and Figure 10A,B). Another relevant factor linked to the inhibitory effectiveness concerns the steric restraints at C2 position. In fact, a significant improvement in the inhibitory potency was observed upon inversion of the C2 configuration, as it occurs going from the epicatechin scaffold (*cis* diastereoisomer) to the catechin scaffold (*trans* diastereoisomer) (Figure 1). A decrease of approximately one order of magnitude was observed in *K_i_*^*^ values for all the substrates, comparing ECG, but also EGCG [21,27], with CG and GCG (Table 1).

Similar to what was previously observed for EGCG [21,27], the inhibition of L-idose reduction by CG, GCG, and ECG appeared as a mixed inhibition type. Different is the case of HNE reduction, which is inhibited by all the inhibitors through an uncompetitive model of action. The same uncompetitive inhibition is exerted by CG and GCG on the GSHNE reduction; this may explain the superimposable effect of these inhibitors on HNE and GSHNE reduction (Figure 2A,B). On the other hand, ECG, which inhibits GSHNE reduction through a mixed inhibition type, resulted in preferentially affecting GSHNE, even though slightly, with respect to HNE (Figure 2C).

It is worth noting that, despite the marked increase of the inhibitory effectiveness of CG and GCG towards L-idose reduction, the parallel increase of the inhibitory power for HNE and GSHNE reduction did not allow any significant enhancement of the differential inhibitory action of these molecules with respect to EGCG [21]. Since comparable *K_i_*^*^ values were observed for the tested inhibitors toward different substrates (Table 1), the differential inhibition of the enzyme by catechins could rely on the different model of action of these molecules. In this regard, the results shown in Figure 9 support the potential of GCG to form binary complexes, starting from different conformations compatible with an interactive pattern with either a sugar substrate (“closed specificity pocket”) or a hydrophobic substrate (“open specificity pocket”). Thus, despite displaying different stability, different EI complexes may be generated starting from different enzyme conformations, which are in turn affected by different substrates. These data acknowledged the high adaptability of AKR1B1 in interacting with quite different molecules, acting both as substrates and as inhibitors [9,10,11,12,41,42,43]. It is, however, hard to predict whether the lower stability of the EI complex built from an “open specificity pocket” conformation of the enzyme (Figure 9B) with respect to the binary complex coming from a “closed specificity pocket” conformation (Figure 9A) is sufficient to rule out its generation, thus giving a rationale to the uncompetitive action of GCG and CG towards HNE and GSHNE reduction. However, the generation of different EI complexes originated from enzyme conformations induced by different substrates would require the occurrence of inductive phenomena generated by the reversible binding of the specific substrate, associated with a slow structural rearrangement of the free enzyme. The same effect could also be caused by the presence of multiple conformations of the free enzyme which involve rearrangement(s) of the binding pocket. In this case, the favorable access for the inhibitor to a specific conformation should lead to a complex so unstable to be kinetically ineffective. With the same approach, the evolution of the ES complex to products may leave the enzyme in a slowly transient conformation to be specifically targeted by the inhibitor. In this latter case, however, the model will be complicated by the marked structural rearrangement(s) occurring on AKR1B1 due to the turnover of the pyridine cofactor [44]. An alternative, easier explanation for the different models of the inhibitory action toward different substrates may relate to the higher affinity of HNE and GSHNE towards AKR1B1 with respect to L-idose. In fact, the ratios K_M(L-idose)_/K_M(HNE)_ and K_M(L-idose)_/K_M(GSHNE)_ were approximately 84 and 36, respectively. On this basis, the GCG-AKR1B1 and CG-AKR1B1 complexes, irrespective of their peculiar conformational assets, may be differently affected by different substrates. Thus, higher affinity substrates (i.e., HNE and GSHNE) would favor the transition of EI toward the ternary complexes, up to an eventual disappearance of the EI complexes. This effect could hardly be conceived in the case of L-idose, the lower affinity substrate, thus leaving the EI complexes present. This reading view of experimental data is in line with the non-classical approach of enzyme inhibition, in which the substrate may generate the ternary complex by direct interaction with the enzyme–inhibitor binary complex [27]. 

In any case, it is conceivable that, as predicted by the classical approach to enzyme inhibition, the inhibitor(s) may bind the ES complex to generate EIS through the preferential interactive trail that the specific substrate may induce, as suggested by the stability of the ternary complexes of GCG and CG with all the substrates (Figure 8 and Figure 10). It is obvious that the two approaches are not mutually exclusive. The presented data, however, are not suitable in supporting the choice for the most favorable model of action, nor in suggesting other possible explanations of the phenomenon.

As noticed above, peculiar evidence appearing from inhibition kinetic data (Figure 4, Figure 5 and Figure 6) concerns the negative slopes occurring in the *^app^K_M_*/*^app^k_cat_* versus [I] related to all the tested catechins acting on HNE reduction and for inhibition of GSHNE reduction by CG and GCG. These results can be fully explained by the occurrence of an incomplete inhibition of the reactions combined with an uncompetitive model of action, two conditions indicated to be necessary to observe the phenomenon [27]. This restriction can be verified, for instance, in the case of the inhibition of GSHNE reduction by ECG (Figure 6C and Table 1). Here, the incomplete inhibitory action of this compound (*k*_+4_ approximately 14% of *k*_+2_) and the evident mixed type of inhibition (*K_i_/K_i_*^*^ approximately 2) did not generate any decrease of the *^app^K_M_*/*^app^k_cat_* versus [I]. In this case, it must be assumed that the driving force for the evolution of the enzyme–substrate complexes (i.e., ES and EIS) towards products is counteracted by the relative stability of EI complex. Finally, no special comment is deserved for those cases where the inhibition was complete, in which the *^app^K_M_*/*^app^k_cat_* versus [I] either may increase (mixed inhibition) or remain constant (uncompetitive inhibition). This is the case of the mixed inhibition exerted by all the catechins/epicatechins on L-idose reduction (*Panel A* of Figure 4, Figure 5 and Figure 6 and Table 1) and by the above-mentioned inhibition of ECG on GSHNE reduction (Figure 6C and Table 1).

The possibility that a substrate may affect not only the model of action of an inhibitor, but also the degree of incompleteness of the inhibitory action may contribute to differential inhibitory phenomena. 

## 5. Conclusions

The results from this study, while offering a characterization of catechin derivatives as powerful inhibitors of AKR1B1, confirm the potential of some molecules in displaying a differential inhibition of the enzyme depending on the substrate under transformation. Our data shed light on some inhibitory features that molecules must exploit to exert differential inhibition; however, the observed differential action remains phenomenological evidence, being structural restrictions for an ARI to become an ARDI still waiting to be clearly defined. In this regard, work is in progress to evaluate the influence of the incomplete inhibitory action exerted by the hemiacetal form of the sugars [45] on the differential inhibition. 

## Figures and Tables

**Figure 1 biology-11-01324-f001:**
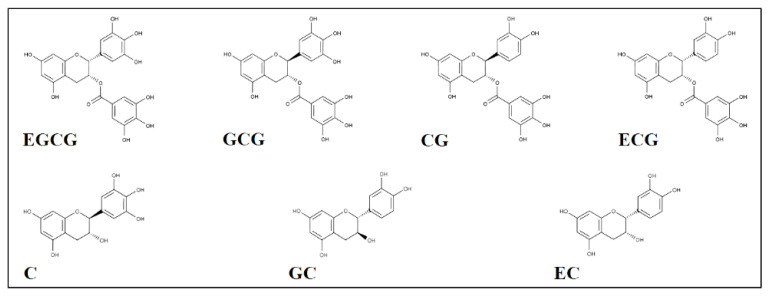
Structure of catechins. EGCG: epigallocatechin gallate; GCG: gallocatechin gallate; CG: catechin gallate; ECG: epicatechin gallate; C: catechin; GC: gallocatechin; EC: epicatechin.

**Figure 2 biology-11-01324-f002:**
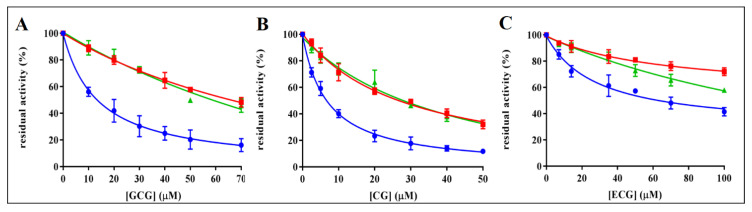
Inhibition curves of AKR1B1 by gallocatechin gallate (GCG), catechin gallate (CG), and epicatechin gallate (ECG). *Panel* (**A**–**C**) refer to GCG, CG, and ECG, respectively. The dose-dependent inhibitory effect was evaluated using 10 mU of AKR1B1 and 8 mM l-idose (blue circles), 0.03 mM HNE (red squares), or 0.045 mM GSHNE (green triangles) as substrate. Error bars (when not visible are within the symbol size) represent the standard deviation of the mean from three to five independent measurements.

**Figure 3 biology-11-01324-f003:**
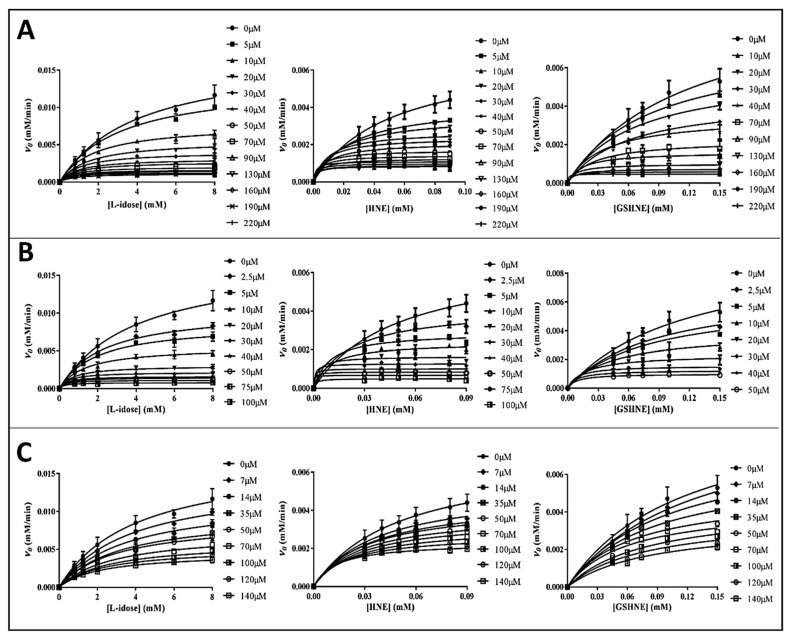
Rate measurements of the AKR1B1 dependent reduction of L-idose, HNE, and GSHNE at different substrates and inhibitors concentration (indicated in legends). *Panel* (**A**–**C**) refers to GCG, CG, and ECG, respectively. The *v*_0_ values (obtained using 10 mU of AKR1B1) versus substrate concentration were interpolated by non-linear regression analysis using the Michaelis-Menten equation with Graphpad Software 7.04. Rate measurements coming from at least three independent measurements are reported as the mean ± SD; bars when not visible are within the symbols size.

**Figure 4 biology-11-01324-f004:**
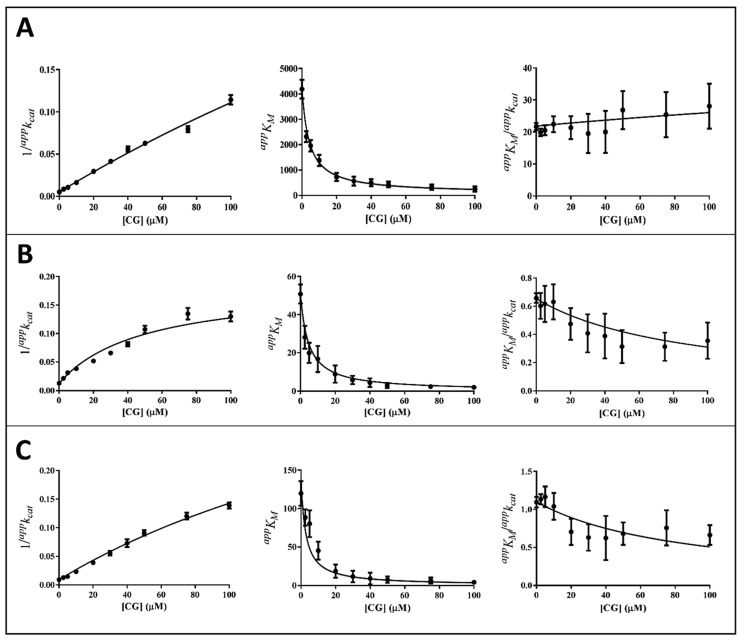
Kinetic analysis of CG inhibition of the AKR1B1-dependent reduction of different substrates. *Panel* (**A**–**C**) refer to L-idose, HNE, and GSHNE, respectively. The secondary plots of apparent kinetic parameters at different GCG concentrations derived from the non-linear regression Michaelis–Menten analysis of primary kinetic data reported in Figure 3B. Bars (when not visible are within the symbols size) represent the standard error of the measurements. The parameters 1/*^app^k_cat_*, *^app^K_M_*, and *^app^K_M_*/*^app^k_cat_* versus [I] were interpolated by non-linear regression using Graphpad Software 7.04. See the text for details.

**Figure 5 biology-11-01324-f005:**
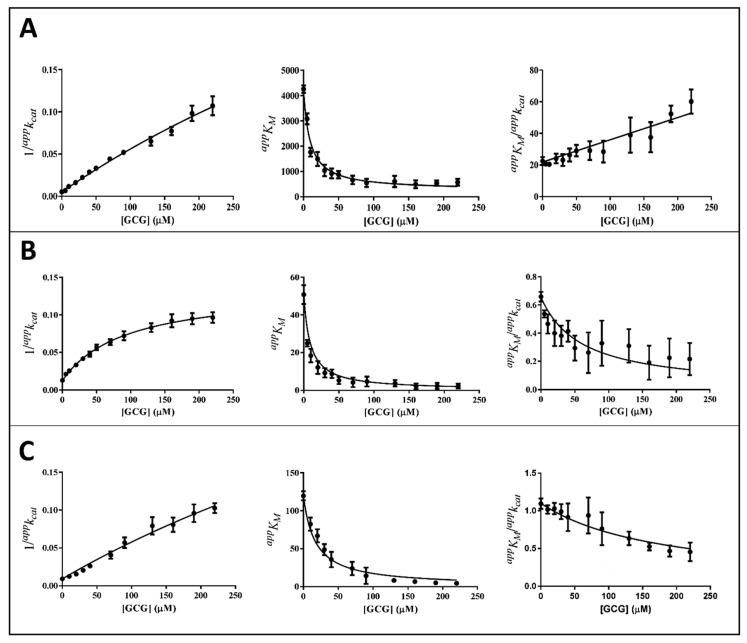
Kinetic analysis of GCG inhibition of the AKR1B1-dependent reduction of different substrates. *Panel* (**A**–**C**) refer to L-idose, HNE, and GSHNE, respectively. The secondary plots of apparent kinetic parameters at different [I] derived from the non-linear regression Michaelis–Menten analysis of primary kinetic data reported in Figure 3A. Bars (when not visible are within the symbols size) represent the standard error of the measurements. The parameters 1/*^app^k_cat_*, *^app^K_M_*, and *^app^K_M_*/*^app^k_cat_* versus [I] were interpolated by non-linear regression using Graphpad Software 7.04. See the text for details.

**Figure 6 biology-11-01324-f006:**
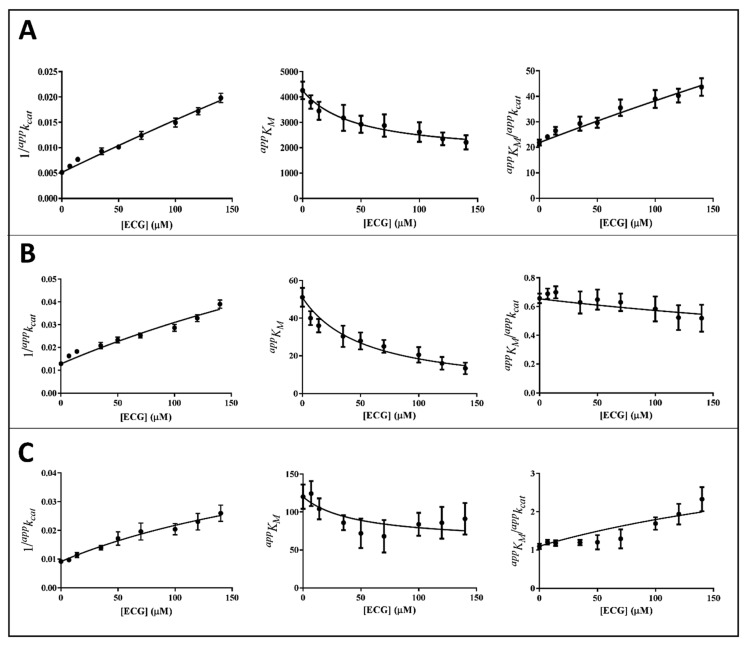
Kinetic analysis of ECG inhibition of the AKR1B1-dependent reduction of different substrates. *Panel* (**A**–**C**) refer to L-idose, HNE, and GSHNE, respectively. The secondary plots of apparent kinetic parameters at different [I] derived from the non-linear regression Michaelis-Menten analysis of primary kinetic data reported in Figure 3C. Bars (when not visible are within the symbols size) represent the standard error of the measurements. The parameters 1/*^app^k_cat_*, *^app^K_M_*, and *^app^K_M_*/*^app^k_cat_* versus [I] were interpolated by non-linear regression using Graphpad Software 7.04. See the text for details.

**Figure 7 biology-11-01324-f007:**
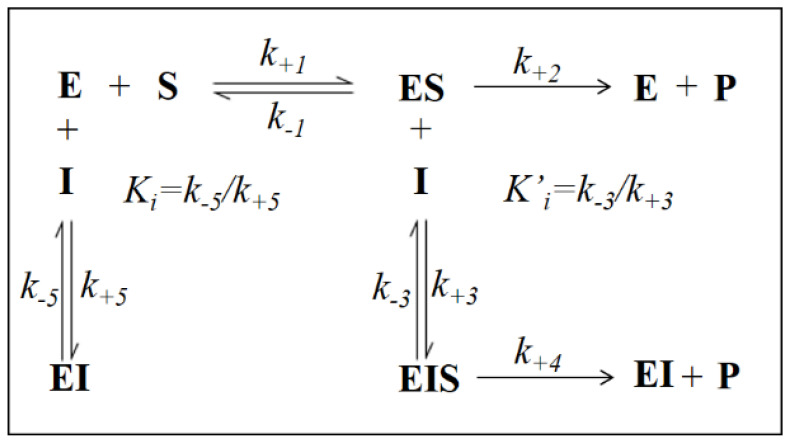
Classical model of incomplete enzyme inhibition when *k*_+2_ > *k*_+4_ > 0. E: enzyme; S: substrate; P: product; I: inhibitor; ES: enzyme-substrate complex; EI: enzyme-inhibitor complex; EIS: enzyme-inhibitor-substrate complex; *k*_+1_, *k*_−1_, *k*_+2_, *k*_+3_, *k*_−3_, *k*_+4_, *k*_+5_, *k*_−5_ are the kinetic constants of the indicated reactions.

**Figure 8 biology-11-01324-f008:**
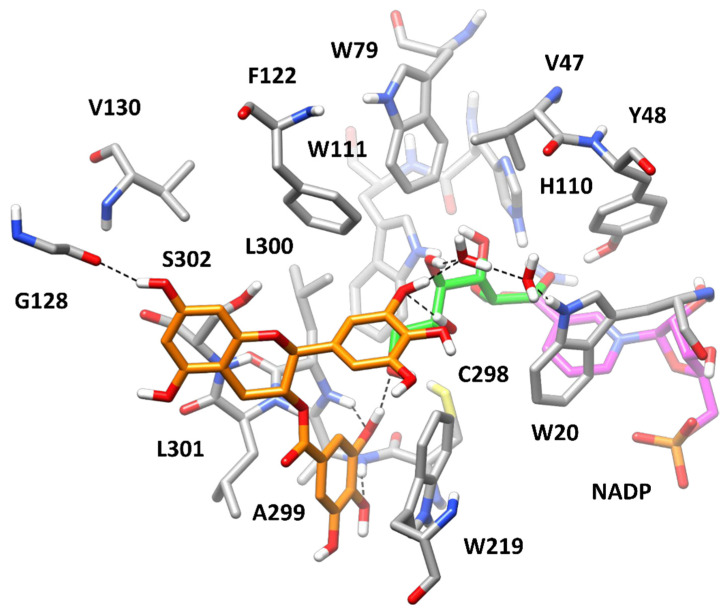
Minimized average structure of AKR1B1 in complex with GCG and L-idose. GCG is shown in orange, the portion of the cofactor adjacent to the ligand binding site is shown in magenta, and the substrate is shown in green. Relevant protein residues (shown in grey) are indicated. Ligand–protein and ligand–substrate H-bonds are shown as black dashed lines.

**Figure 9 biology-11-01324-f009:**
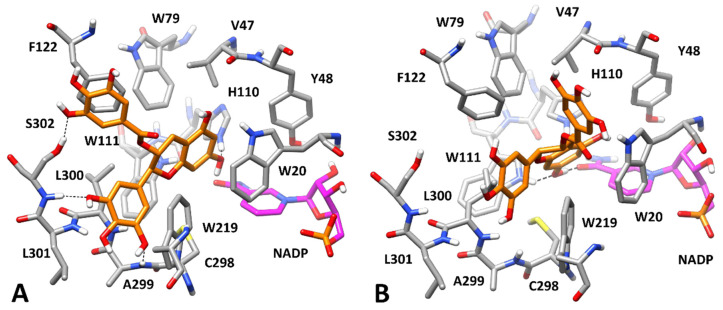
Minimized average structure of AKR1B1 complexed with GCG. *Panel* (**A**,**B**) refers to the AKR1B1 specificity pocket in the closed and open conformation, respectively (see text for details). GCG is shown in orange and the portion of the cofactor adjacent to the ligand binding site is shown in magenta. Relevant protein residues (shown in grey) are indicated. Ligand–protein H-bonds are shown as black dashed lines.

**Figure 10 biology-11-01324-f010:**
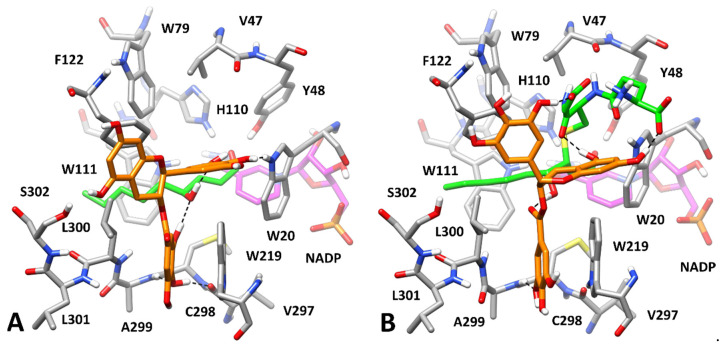
Minimized average structure of AKR1B1 bound to HNE and GSHNE, in complex with GCG. *Panel* (**A**,**B**) refer to HNE and GSHNE, respectively. GCG is shown in orange, the portion of the cofactor adjacent to the ligand binding site is shown in magenta, and the substrates are shown in green. Relevant protein residues (shown in grey) are indicated. Ligand–protein and ligand–substrate H-bonds are shown as black dashed lines.

**Table 1 biology-11-01324-t001:** Inhibition constants of CG, GCG, and ECG for l-idose, HNE, and GSHNE reduction. The values of the constants, derived from the non-linear regression analysis of 1/*^app^k_cat_*, *^app^K_M_*, and *^app^K_M_*/*^app^k_cat_* versus [I] (Figure 4, Figure 5 and Figure 6) through Equations (1)–(4). *K_i,_* and *K^*^_i_* are expressed in µM; *k*_+4_ is expressed in min^−1^. The *K_M_* values for L-idose, HNE, and GSHNE in the absence of inhibitors were assumed 4260 µM, 51 µM, and 120 µM, respectively. Data are reported as the mean ± SD; ND: not determined. ^a^ Calculated using *k*_+2_ values of 195 ± 6 min^−1^, 78 ± 4 min^−1^, and 109 ± 8 min^−1^ for L-idose, HNE, GSHNE, respectively.

Catechin	L-Idose	HNE	GSHNE
**CG**			
*K_i_*	246 ± 170	ND > 10^21^	ND > 10^22^
*K_i_^*^*	4.1 ± 0.2	4.5 ± 0.4	5.0 ± 0.2
*k* _+4_	1.4 ± 0.6	4.7 ± 0.7	4.1 ± 0.8
*k*_+4_/*k*_+2_ (%) ^a^	0.7 ± 0.3	6.0 ± 0.9	3.8 ± 0.8
**GCG**			
*K_i_*	71 ± 9	ND > 10^15^	ND > 10^23^
*K_i_^*^*	7.1 ± 0.4	9.0 ± 0.5	17 ± 2
*k* _+4_	5.6 ± 0.5	9.5 ± 0.9	6.0 ± 1.0
*k*_+4_/*k*_+2_ (%) ^a^	2.9 ± 0.25	12.2 ± 1.1	5.5 ± 0.9
**ECG**			
*K_i_*	116 ± 8	ND > 10^4^	91 ± 10
*K_i_^*^*	46 ± 5	54 ± 9	46 ± 11
*k* _+4_	5.2 ± 3.2	6.7 ± 2.7	15.6 ± 4.3
*k*_+4_/*k*_+2_ (%) ^a^	2.7 ± 1.7	8.6 ± 0.4	14.3 ± 3.9

## Data Availability

The data presented in this study are available in the article and in Appendix A.

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
