# Peer review of "Dissecting the Activity of Catechins as Incomplete Aldose Reductase Differential Inhibitors through Kinetic and Computational Approaches"

_biology, 2022, doi:10.3390/biology11091324_

Round 1
Reviewer 1 Report
The work described herein reports the characterization of different catechin derivatives as aldose reductase differential inhibitors. However, the authors should take under consideration the following points.
1. In the Introduction section, the mechanism of diabetic complications and reason to develop ARDIs should be clarified more clearly.
2. In determination of AKR1B1 activity, the positive control should be added.
3. Authors used catechin derivatives as ARDIs candidates, however, the structural characteristics of all the derivatives should be determined first of all, such as NMR, purity, HRMS and so on. This is very important to other following research in this paper.
4. Check language and format-editing thoroughly.
Author Response
- In the Introduction section, the mechanism of diabetic complications and reason to develop ARDIs should be clarified more clearly. We have modified the Introduction, inserting some considerations which, we hope, should clarify the point raised by the Reviewer.
- In determination of AKR1B1 activity, the positive control should be added. The description of the positive control, i.e the assay in the absence of inhibitors and in the presence of the solvent, was erroneously omitted in the original version. We apologize for this. We have now inserted this description in section 2.2.
- Authors used catechin derivatives as ARDIs candidates, however, the structural characteristics of all the derivatives should be determined first of all, such as NMR, purity, HRMS and so on. This is very important to other following research in this paper. All the compounds tested as inhibitors were from commercial source. The purity of each compound, as determined by the Producer, is now reported in Section 2.1.
- Check language and format-editing thoroughly. A check for language and format has been performed.
Reviewer 2 Report
The manuscript submitted by Balestri et al. belongs to a series of interesting research reports by Del Corso’s group, who coined the concept of “differential inhibition” applied to the search of selective inhibitors against aldose reductase (AKR1B1). AKR1B1 is involved in secondary diabetic complications, and its inhibition has been long sought-after in order to block glucose conversion (in the polyol pathway) and 3-glutathionyl-4-hydroxynonenal (GSHNE) reduction (during inflammatory response). AKR1B1 also exerts an important detoxifying role by reducing a number of cytotoxic aldehydes, and thus complete inhibition of its enzymatic activity is not desirable. The rationale behind the use of differential inhibitors is to block the transformation of glucose and GSHNE, with a null or minimum impact on the removal of toxic aldehydes. In the present study, the authors extended previous reports and used both a kinetic and a computational approach to analyze the effect of additional catechin derivatives as differential inhibitors: gallocatechin gallate (GCG), catechin gallate (CG) and epicatechin gallate (ECG). L‑idose, 4-hydroxynonenal (HNE) and GSHNE were used as substrates.
The manuscript is clearly written and the experimental work was carefully designed and performed. The authors offer a convincing explanation for the inhibition type and the degree of incompleteness of the inhibitory action, both factors related to the substrate type and contributing to differential inhibition. However, the precise structural features for an aldose reductase inhibitor to become a differential inhibitor still remain to be elucidated at this time.
Minor points:
- The title should be more informative (i.e. to state the kinetic and computational methods used) in order to differentiate the present work form previous ones that also used catechins, such as Ref. 21.
- Line 181 and ff: Please define “CLs”.
- Line 276: Please explain how the starting values were imposed and where these imposed values come from.
- Line 320: Please define RMSD. Is this value referred to the ligand coordinates after MD with respect to the ligand disposition after molecular docking?
- Line 394 and ff: Please use accepted nomenclature for the "specificity pocket" in the field, according to Ref. 40. “Selectivity pocket” and “specificity pocket” are used interchangeably throughout the text. Do they refer to the same pocket?
- Line 416 and ff: Binding free energy values are negative values. Thus when we consider a “stronger” binding (with lower energy), the value should be lower (more negative) rather than higher (less negative). Please consider revising lines 431, 496, etc., and exchange “lower” and “higher” accordingly.
Author Response
The title should be more informative (i.e. to state the kinetic and computational methods used) in order to differentiate the present work form previous ones that also used catechins, such as Ref. 21. We thank the Reviewer for the suggestion; we have modified the title, which now mentions both the adopted experimental approaches and the observed incomplete inhibition: “Dissecting the activity of catechins as incomplete aldose reductase differential inhibitors through kinetic and computational approaches”.
Line 181 and ff: Please define “CLs”. We apologize for the omission. In response also to Rev.#1, we have described the obtainment of IC50 values with more details (Section 2.2), also inserting the definition of the confidence limits (CLs) of the measurements.
Line 276: Please explain how the starting values were imposed and where these imposed values come from. We thank the Reviewer for the observation. The starting values of kcat and KM come from at least 20 independent experiments in which the kinetic parameters in the absence of inhibitors were determined. This is now stated in the revised version.
Line 320: Please define RMSD. Is this value referred to the ligand coordinates after MD with respect to the ligand disposition after molecular docking? The root-mean-squared deviation (RMSD) refers to the ligand coordinate during the MD with respect to its initial docking pose. The RMSD is calculated during the whole simulation and then averaged. In order to make this clear to the reader, the sentence the Reviewer is referring to (pag. 13 of revised version) has been rephrased.
Line 394 and ff: Please use accepted nomenclature for the "specificity pocket" in the field, according to Ref. 40. “Selectivity pocket” and “specificity pocket” are used interchangeably throughout the text. Do they refer to the same pocket? The two terms refer to the same pocket, so we have modified the text in order to use only the original “specificity pocket” term.
Line 416 and ff: Binding free energy values are negative values. Thus when we consider a “stronger” binding (with lower energy), the value should be lower (more negative) rather than higher (less negative). Please consider revising lines 431, 496, etc., and exchange “lower” and “higher” accordingly. We have modified the text to avoid misunderstanding. We apologize for this.
Reviewer 3 Report
Balestri et al. report on the differential inhibition of the human aldose reductase by the catechins gallocatechin gallate (GCG) and catechin gallate (CG). Aldose reductase (AKR1B1), an NADPH-dependent oxidoreductase can directly or indirectly generate cell damage when entering into glucose metabolism, especially in hyperglycemic conditions. At the same, AKR1B1 can also have a detoxifying role in reducing a number of cytotoxic aldehydes. Thus, identification of a differential inhibitor, one that inhibits an enzyme based on the substrate, for AKR1B1 is paramount. Both GCG and CG preferentially affected aldoses and 3-glutathionyl-4-hydroxynonenal reduction with respect to 4-hydroxynonenal reduction. The kinetic results and structure modelling indicate that the substate not only affects the mode of action of an inhibitor, but can also affect the degree of incompleteness of the inhibitory action.
Specific Comments:
(1) A few grammatical and typographical errors need to be resolved.
(2) Are Figures 2-6 necessary in the manuscript? Would the supplemental section be a better fit?
Author Response
(1) A few grammatical and typographical errors need to be resolved. We have checked the text for grammatical and typo errors and made proper corrections.
(2) Are Figures 2-6 necessary in the manuscript? Would the supplemental section be a better fit? Concerning the proper location of Figs. 2-6, we believe that they deserve the current location in the main text. The data reported in figs. 2-6 represent the primary data obtained through the kinetic approach and give a direct information of the quality and also of the limits of the measurements and of their fitting with proper equations. In any case we thank the Reviewer for the suggestion
Round 2
Reviewer 1 Report
The author have revised manuscript based on the comments of the reviewer, and there were no other comments.